# Chronic Aflatoxin Exposure and Cognitive and Language Development in Young Children of Bangladesh: A Longitudinal Study

**DOI:** 10.3390/toxins14120855

**Published:** 2022-12-03

**Authors:** Mustafa Mahfuz, Md. Shabab Hossain, Md. Ashraful Alam, Md. Amran Gazi, Shah Mohammad Fahim, Baitun Nahar, Tahmeed Ahmed

**Affiliations:** 1Nutrition and Clinical Services Division (NCSD), International Centre for Diarrhoeal Disease Research, Bangladesh (icddr,b), Mohakhali, Dhaka 1212, Bangladesh; 2Faculty of Medicine and Health Technology, University of Tampere, 3352 Tampere, Finland

**Keywords:** aflatoxin, child development, cognitive, Bayley

## Abstract

Aflatoxin can cross the blood–brain barrier, damage brain tissues, and have the potential to harm the development of the human brain. Although dietary aflatoxin exposure is common in children, there is a paucity of data on aflatoxin exposure and child developmental outcomes. The child’s cognitive, motor, and language functions were assessed using the Bayley Scales of Infant and Toddler Development-III or BSID-III at the same time points. Association between exposure to aflatoxin and subtests of BSID-III were examined using mixed-effect linear regression. Aflatoxin assays were performed on 194, 167, and 163 children at 15, 24, and 36 months of age, and chronic aflatoxin exposure was detected in 20.6%, 16.8%, and 60.7% of children, respectively. Multi-variable analyses showed that aflatoxin exposure was independently related to the children’s cognitive score (β: −0.69; 95% CI: −1.36, −0.02), receptive language score (β: −0.90; 95% CI: −1.62, −0.17), and expressive language score (β: −1.01; 95% CI: −1.96, −0.05). We did not observe any association between exposure to aflatoxin and the motor function of children. Chronic exposure to aflatoxin exposure was linked to reduced cognitive, expressive, and receptive language scores of the study children. Further research is needed in a different setting to confirm this novel finding.

## 1. Introduction

Exposure to dietary aflatoxin is a major public health issue, especially because of its association with the pathogenesis of liver cancer [1]. Aflatoxins are subordinate metabolites of the fungus *Aspergillus flavus* and *Aspergillus parasiticus*, which are naturally occurring contaminants of regularly consumed food items, for instance, maize, groundnut, spices, rice, nuts, dried food, and flour [2]. About 4.5 billion people are exposed to an unknown amount of aflatoxin through direct consumption and also indirectly through animals from contaminated feeding across the globe. The hot and humid climate in tropical and sub-tropical regions, floods, and poor storage result in the enhanced growth of mould and the ensuing production of toxins. The exposure is most prevalent in Sub-Saharan Africa and some countries of Southeast Asia. Food insecurity, lack of dietary diversity, poor agricultural practices, particularly during pre- and post-harvest seasons, limited or no awareness of aflatoxin, and the lack of enforcement of regulatory limits are the other main contributing factors associated with high-level exposure in this population [3].

Aflatoxin B1 (AFB1) is the most common and most toxic among different types of aflatoxins [4]. The aflatoxin B1-lysin albumin adduct (AFB1-lys) is a sensitive and commonly used biomarker for the detection of chronic aflatoxin exposure [5]. In high prevalent areas of East and West Africa, where the intake of maize and groundnut is high, the detection of AFB1 was found to be as high as 95% in the blood samples [6]. Recent studies carried out in Tanzania, Nepal, and Bangladesh revealed that a high proportion of samples from children had detectable AFB1-lys [7]. Another study in Nepal and Bangladesh found AFB1-lys in the plasma samples of mothers, newborns, and the same children at 2 years of age, while 100% of the cord blood of newborns had detectable levels of aflatoxin whose mothers were exposed to aflatoxin, suggesting vertical transmission [8]. A child can be exposed to aflatoxin in utero, through breast milk, and from complementary feeding after weaning [9].

Epidemiological studies conducted in Sub-Saharan Africa and elsewhere have observed a significant association between long-term dietary aflatoxin exposure and child growth deficit including childhood stunting [10]. However, recent studies could not establish any association between AFB1-lys detection in plasma samples with child growth and suggested the presence of a threshold of exposure, indicating that values above the threshold might be responsible for growth faltering [11]. Childhood stunting is associated with the delayed cognitive development of children [12]. Currently, at least 250 million children under the age of five years in low- and middle-income countries (LMICs) are vulnerable to reaching their full developmental potentials [13]. However, no data have explored the relationship between aflatoxin exposure and childhood development.

Aflatoxins are highly lipid-soluble and are easily taken in the intestine after ingestion and enter the circulation [14]. It is then metabolised in the liver by the cytochrome p450 enzyme system and is converted into highly reactive exo-epoxide that binds with albumin to form albumin adducts [15]. It also binds with guanine to form DNA adducts and can initiate carcinogenesis [16]. Although the reactive epoxide primarily affects the liver, it can also affect other organs and cells including lymphocytes, gut, lungs, and the brain [17]. Aflatoxin may cause growth deficits in children by damaging enterocytes, leading to poor absorption of essential nutrients. It also induces systemic immune activation and reduces insulin-like growth factors (IGF) through liver toxicity [18]. Moreover, aflatoxin exposure has been reported to produce biochemical alterations, leading to injury to brain tissues in both human and animal studies [17,19]. The direct consumption of AFB1 developed brain injury in rat models [20] and had a neurotoxic effect on the cerebellar cortex [21]. Its prolonged consumption can upset the hypothalamus in regulating neuropeptides that control feeding behaviour [22]. Aflatoxin was found to be present in 81% of brain autopsies in children from communities with high levels of food contamination with aflatoxin [23]. Another study demonstrated that aflatoxins could affect the integrity of the blood–brain barrier and pass through it [17]. In this study, human brain microvascular endothelial cells were grown in tissue culture flasks and exposed to AFB1 at different concentrations. This study showed that the constituents of the blood–brain barrier, the microvascular endothelial cells, were susceptible to the cytotoxic effect of aflatoxin. They also showed that even at lower concentrations, aflatoxin can damage these primary cell lines [17]. Another study examined the histopathological effect of AFB1 and its withdrawal in a rat model and observed that AFB1 caused decreased astrocytes in the cerebral cortex, increased astrocytes in hippocampus, and decreased the total cell and neuronal number in the hippocampus; all of these changes were reversed after the withdrawal of AFB1 exposure [19].

The studies described above indicate that chronic exposure to aflatoxin has the potential to invade and damage brain tissues in infants and young children during the crucial period of child development. However, no studies have been conducted to explore whether early aflatoxin exposure compromises the cognitive and language functions of young children. This current analysis takes the opportunity of the MAL-ED aflatoxin study, a companion study of the Etiology, Risk Factors, and Interactions of Enteric Infections and Malnutrition and the Consequences for Child Health and Development (MAL-ED) birth cohort study in Bangladesh to explore the relationship between chronic aflatoxin exposure and cognitive and language development of children aged between 15 months and 36 months.

## 2. Results

Data on aflatoxin exposure and cognitive, language, and motor functions were available for analysis for 194, 167, and 163 Bangladeshi children enrolled in the MAL-ED aflatoxin study at 15, 24, and 36 months of age, respectively. Baseline data revealed 51% of children to be female, with an average birth weight of 2.82 kg, and median birth order was 2. The mothers’ median age was 25 years, having a median of 5 years of schooling. The median maternal Raven’s factor was 13. Mean HOME emotional and verbal responsivity score, HOME environmental safety score, and HOME cleanliness score were 11.2, 3.2, and 3.9, respectively. Incidence of diarrhoea and acute lower respiratory tract infection (ALRI) were 3.3 and 0.4 episodes per child per year, respectively. The median household income was USD100 (Table 1).

Aflatoxin assays were performed for 194, 167, and 163 children in months 15, 24, and 36, respectively. Chronic aflatoxin exposure was detected in 20.6%, 16.8%, and 60.7% of children at 15, 24, and 36 months, respectively. The average length-for-age z-scores (LAZ) at 15, 24, and 36 months were −1.8, −2.0, and −1.9, respectively. The cognitive and motor T-scores calculated using BSID-3 performed at 15, 24, and 36 months is presented in Table 2. The distribution T-scores in children measured at 15, 24 and 36 months is presented in Figure 1.

After adjusting for age and sex, the bivariate analyses for cognitive, language, and motor functions of children between 15 months and 36 months with different explanatory variables are presented in Table 3. Bivariate analyses showed that cognitive scores were inversely related to aflatoxin exposure, with positive associations with birth weight, haemoglobin concentration, maternal education, and monthly household income. Receptive language function was negatively associated with aflatoxin exposure and positively associated with birth weight, LAZ, incidence of diarrhoea and monthly income. Expressive language function was positively associated with birth weight, LAZ, haemoglobin, and maternal Raven’s factor. The fine motor function of children was associated with birth weight, maternal education, Raven’s factor, and HOME cleanliness score and the gross motor function was associated with birth weight, haemoglobin, maternal education, and HOME environmental safety score.

Multi-variable analyses showed that the children’s cognitive function was independently associated with aflatoxin exposure, age in months, and Raven’s factor (Table 4). Receptive language score was associated with aflatoxin exposure, age, birth weight, and Raven’s factor. Expressive language score was associated with aflatoxin exposure, age, and birth weight of the children. No association between aflatoxin exposure and motor function of the children was found. The fine motor function of children was associated with the children’s age, Raven’s factor, and HOME cleanliness. Gross motor function of children was associated with children’s age, LAZ, the incidence of ALRI, and Raven’s factor (Table 4).

## 3. Discussion

Our analysis showed that aflatoxin exposure was associated with reduced cognitive scores and both the expressive and receptive language scores of the study children. To our knowledge, this is the first report where the effect of aflatoxin exposure was tested with child development outcomes, making the finding a novel one. Cognitive function, as evaluated by cognitive scores, represents the higher mental ability of children including problem-solving capacity, counting, working memory, and ability to cope with the environment. In the multi-variable analysis, we observed that aflatoxin exposure was negatively associated with cognitive scores. It is already established that chronic aflatoxin exposure can cross the blood–brain barrier, invade brain tissues, make structural and biochemical changes, and have all the potential to harm infants and young children during the crucial period of child development [17], which explains the negative association with the cognitive scores. The multi-variable analysis also revealed that age, birth weight, and maternal intelligence were positively associated with the cognitive function of the children. Recently, a systematic review and meta-analyses showed that in low- and middle-income countries (LMICs), the extent of early life relationships affected the development of children [24]. The review examined data collected from a total of twenty-one studies covering 20,882 children from thirteen countries and assessed the correlations between exposure to different risk factors and child development. They found higher maternal education to be associated with higher cognitive scores, while low birth weight, pre-term birth, maternal short stature, childhood anaemia, and access to clean water and sanitation were found to be associated with lower cognitive and motor scores.

Previously, studies have shown chronic aflatoxin exposure to be associated with child growth impairment, particularly with stunting [25,26]. However, several recent studies from Asia including Bangladesh found no association of aflatoxin exposure with child growth, despite a substantial proportion of children being exposed to aflatoxin [7,27]. Compared to the concentration of aflatoxin metabolites, aflatoxin lysine-adduct or aflatoxin albumin adducts found in plasma samples in studies previously conducted in Africa, the concentrations observed in these recent studies had much lower concentrations of metabolites. Therefore, researchers have assumed that exposure to a low concentration of aflatoxin lysine adducts in plasma samples could not cause growth faltering in children. Perhaps chronic exposure to aflatoxin in a low-concentration cannot cause linear growth faltering in young children, and might impair cognitive and language development.

There were some limitations in this study. Aflatoxin exposure in children at 2–3 months of age or at birth was not tested as blood samples were not collected during this period. Therefore, cognitive functions at around 6–7 months could not be assessed in relation to early aflatoxin exposure. The unavailability of maternal aflatoxin exposure was another limiting factor to explain whether the exposure was exclusively dietary sourced from family food or through breast milk. Aflatoxin exposure data were also missing in several timepoints as some participants did not provide samples at different time points. Moreover, the aflatoxin concentration in common food items was not measured.

Failure in achieving optimal development in early life has long-term impacts on education [28] and income generation [29], passively contributing to poverty. This cycle has the potential to cross generations [30]. Children suffering from developmental deficits result in an annual loss of approximately 20% of adult income [31], at an estimated cost of USD 177 billion across the globe for delayed physical growth alone [32]. Recognising the burden and costs related to childhood developmental deficits, the 2030 Sustainable Development Goals (SDGs) directly aim for early childhood development (ECD) to be placed under SDG 4 [33] to ensure quality ECD care and primary education for all children. The association we observed between aflatoxin exposure and lower cognitive, expressive, and receptive language scores has been a new but important finding. However, this needs to be explored in a different population before generalising the results. 

## 4. Materials and Methods

### 4.1. Study Design and Participants

This was a longitudinal observational study where healthy newborns were enrolled and followed post-partum up to 36 months of age. These children were enrolled in the MAL-ED birth cohort study (registration no: NCT02441426) conducted in an informal settlement of Dhaka, Bangladesh where information on child development and chronic aflatoxin exposure data from plasma samples were collected repeatedly when the children were 15, 24, and 36 months old.

### 4.2. Study Site

Detailed information on the study site and settings of the MAL-ED Bangladesh study have already been reported [34]. This study was situated in an urban informal settlement at Bauniabadh, Mirpur, Dhaka. The majority of the people living in the study area belong to low socioeconomic status with sub-optimal sanitary conditions. A well-distinct inclusion and exclusion criteria were utilised during enrolment, where healthy newborns of either sex within 17 days after birth, whose parents provided consent to participate in the study, were included while maternal age less than 16 years, multiple pregnancy, newborns with a very low birth weight (<1.5 kg), a severe disease that required hospitalisation, and children with chronic disease or congenital anomalies were excluded from the study [34]. Enrolment of the participants began in February 2010 and enrolment commenced within 17 days of birth and the newborns were longitudinally followed until 36 months of age. In order to address the seasonal differences, only 10 participants were enrolled each month, and enrolment was completed in February 2012. Blood samples were collected at 7 months, 15 months, 24 months, and 36 months of age. Only children who had available aflatoxin assay results were included in the current analysis. More details on the MAL-ED aflatoxin study have already been reported [8].

### 4.3. Data Collection

This study recorded data on the birth date and weight at birth during enrolment. Moreover, data on anthropometry, initiation, frequency, and types of breastfeeding, complementary feeding, status on maternal schooling and anthropometry, and basic socio-demography, food security, and economic status were collected at the baseline [25,26,27]. In addition to a prospective collection of monthly anthropometry and socio-economic data and half-yearly collection of data on income, asset, household food security, and water sanitation and hygiene-related behaviour [9,25,26], this study also collected exhaustive data on morbidity and dietary consumption through twice-a-week household surveillance [28].

### 4.4. Biological Sample Collection

In this study, all biological samples including plasma samples were collected by trained research staff using a standard protocol [25].

### 4.5. Aflatoxin Plasma Biomarker Assay

Assays for Aflatoxin B1-lysine albumin adduct (AFB1-lys) from plasma samples were conducted at Johns Hopkins University, USA. Using a validated methodology [35], the concentrations of AFB1-lys in the plasma samples were analysed by isotope dilution mass spectrometry (IDMS). In brief, the plasma (200 µL) sample was mixed with pronase (Millipore Catalogue #537088-100 uM) and an internal standard (10 µL × 0.1 ng AFB1-D4-lys/mL), vortexed, and then incubated for 18 h at 37 °C. Ultra-performance liquid chromatography (UPLC) coupled with mass spectrometry was used to analyse the eluent. A concentration of 0.5 pg of AFB1-lys in per mg of albumin was considered as the limit of detection and three samples were run every day for quality control purposes [8].

### 4.6. Child Development

To assess global child development, the third version of the Bayley Scales of Infant and Toddler Development (BSID-III) (Bayley 2005) was used. This tool is widely used for children aged 1–42 months and the assessments take between 45 and 60 min to administer. It contains three individually administered subscales tested with children: cognitive, language, and motor subscales. It also includes two rating scales completed by parents or caregivers consisting of the social-emotional scale and adaptive behaviour scale. The adaptive behaviour questionnaire was not used in the present study [36].

The cognitive scale evaluates the higher-order mental ability of a child that includes thinking and executive functions such as working memory, attention to acquainted and unacquainted objects, thinking skills, familiarity with numbers, counting skills, and learning how to manipulate the surrounding environment.

The motor scale of the BSID-III has a couple of subscales—gross motor and fine motor. Hand/eye coordination, skills for motor control of fingers and hands including pincer grasp, holding a spoon or pencil, drawing, etc., can be measured by the fine motor sub-scale. In contrast, complex movements such as walking, sitting, jumping, etc. can be measured by the gross motor sub-scale.

A child’s ability to understand, communicate, and express feelings can be assessed by the language scale. The receptive communication subset of this scale is used to assess the child’s ability to identify and remember the different sounds, and understand and follow the meaning of spoken words. On the other hand, the expressive subset tests the child’s ability to communicate using gesticulations, or facial expressions, or uttering words, sentences, or phrases.

The social-emotional (SE) scale measures the ability of a child’s social and emotional interactions, control of emotions, development of positive attitudes, and gratifying bonds with others. This developmental skill is measured by using a list of questions, administered to the caregivers or mothers. Questions consist of the social and emotional milestones of children that are usually developed at particular ages and their attention levels to bright and colourful things. This scale also measures how comfortable or difficult it is to gain the child’s attention or soothe the child, fulfil their demands, and identify the way the child use words for communication.

The previous version of the Bayley scale (BSID-II) was validated [36] and already used in several studies in Bangladesh [37,38]. Some studies in Bangladesh have started to use the BSID-III and the published data showed a decent correlation of the subscales with different predictors such as age, morbidity, and growth [39].

In this study, trained psychologists utilised the BSID-III scale to measure cognitive, language, and motor functions of enrolled children at 15 months, 24 months, and 36 months of age. For the language and motor subsets, both expressive and receptive language subsets, and fine and gross motor functions were assessed.

### 4.7. Nutritional Status

As a proxy for the children’s nutritional status, anthropometry was performed by trained research staff. This includes measuring the length and weight of children using standard operative procedures and measuring scales. A Seca 727 Electronic Baby Scale (Seca, Hamburg, Germany) was used to measure the body weight and a Seca 416 length board (Seca, Hamburg, Germany) was used to measure the length of the children. All anthropometry was conducted in a fixed time with minimal clothing and data were converted to z-scores using the new World Health Organisation’s growth standard [40].

### 4.8. Home Environment

To evaluate the quantity and quality of the existing psychosocial home environment at children’s homes, a trained field staff visited the children’s homes and administered the modified version of the Home Observation for Measurement of the Environment (HOME) Inventory (Infant/Toddler version) [41]. The modification of HOME was made for use in Bangladesh by rephrasing some of the questions. This scale consists of six subscales that are again sub-divided into 48 items. Sub-scales are (1) responsivity to which parents and the environment is responsive; (2) acceptance through avoiding punishment and restriction; (3) organisation by providing structure to the child’s living environment; (4) providing proper learning and play materials; (5) involvement of the parents; and (6) variety in regular stimulation to have a balanced experience in life. The total score was calculated by summing up all of the positive responses.

### 4.9. Maternal Measures

The Self Reporting Questionnaire-20 (SRQ-20) was used to quantify the depressive symptoms of mothers by trained research staff [42]. The World Health Organisation has developed this scale to be used by developing countries [43] to measure the maternal psychological adjustments linked to depressive symptoms. This questionnaire comprises 20 polar questions that can be answered with a “yes” or “no” [44]. In the current study, research staff administered this questionnaire to mothers at 6 months, 15 months, and 24 months of age, and the scores were analysed as continuous data. Moreover, the anthropometry of the mothers was performed after two months post-partum and BMI was calculated.

### 4.10. Maternal Intelligence

Maternal intelligence was assessed by administering the Raven’s Coloured Progressive Matrices and Standard Progressive Matrices (RCM) [45]. The RCM is a widely used nonverbal test to measure the general cognitive ability or eductive (drawing out) ability to solve novel reasoning problems and fluid intelligence such as comprehension, problem-solving, and learning. It usually consists of 60 elements and the participants are requested to complete a pattern by identifying missing items in each assessment, and then answers are recorded. The assessment took approximately 20–30 min conducted by the trained testers.

### 4.11. Family Characteristics

In this birth cohort study, using a standard questionnaire, data on socio-economy and demography were collected at enrolment. The particulars of the questionnaire and data collection methods were reported elsewhere [34]. The current paper utilised data on maternal age and education. Maternal year of schooling was further categorised into five categories starting from no schooling to higher secondary level education (grades 11 to 12) and above.

### 4.12. Statistical Analysis

We measured the child development by using five subtests of BSID-III to assess the cognitive, language, and motor developmental status of the study children at three follow-ups conducted during 15, 24, and 36 months of age. In the current analysis, we converted all raw scores to T-scores (i.e., standardising to a mean of 50 and SD of 10). Socio-economic data were described by the mean (±SD) or median with an interquartile range (IQR), as appropriate. We also used a proportion estimate to summarize the categorical variables. To compare the BSID-III subtests by aflatoxin status, we used the box-and-whisker plot. We examined the association between aflatoxin and five subtests of BSID-III including T-scores of cognitive, expressive language, receptive language, gross motor, and fine motor functions of the children using mixed-effect linear regression models including a random intercept for the study child. To adjust for clustering by the child, we included the child as a random effect. We used the variance inflation factor (VIF) to examine for multicollinearity among the independent variables. At first, unadjusted associations between each explanatory variable with the outcome variables were examined by bivariate analysis. Explanatory variables with a *p*-value of less than 0.2 in bivariate analysis were included in the multivariable models. Final models were based on the lowest AIC and BIC values. Statistically significant was defined by a *p*-value of less than 0.05. R software version 3.6.3 was used to perform all data analyses.

### 4.13. Ethics

We collected informed written consent from mothers at the time of enrolment. The Research Review Committee (RRC) and Ethical Review Committee (ERC) of icddr,b had approved the research protocol.

## Figures and Tables

**Figure 1 toxins-14-00855-f001:**
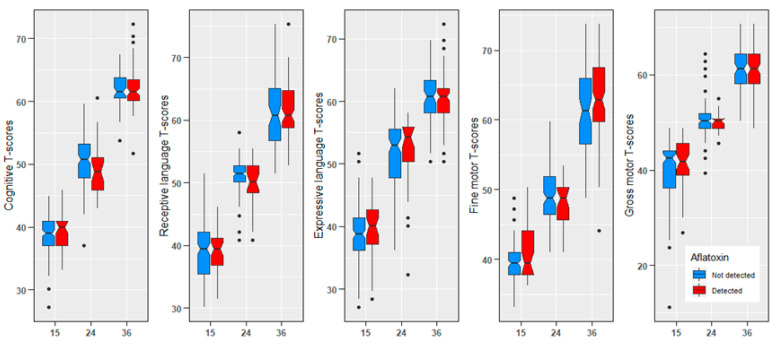
Cognitive, language, and motor T scores of children at 15 months, 24 months, and 36 months of age.

**Table 1 toxins-14-00855-t001:** Baseline characteristics of study participants.

Indicator	Unit	*n* (%), Mean (SD), or Median (IQR)
Female sex, *n* (%)	%	106 (51)
Birth weight, mean (SD)	kg	2.82 (0.41)
Birth order, median (IQR)	order	2 (1, 2)
Mother age, median	year	25 (20, 28)
Maternal education, median (IQR)	year	5 (2, 7)
Maternal height, mean (SD)	cm	148.91 (5.21)
Maternal raven’s factor, median (IQR)	score unit	13 (9, 22)
HOME emotional and verbal responsivity score, mean (SD)	score unit	11.16 (1.3)
HOME environmental safety score, mean (SD)	score unit	3.19 (0.65)
HOME cleanliness score, mean (SD)	score unit	3.98 (0.11)
Monthly household income, median (IQR)	USD	100 (72, 138)
Incidence of diarrhoea, mean (SD)	Number of new cases in a year	3.29 (2.12)
Incidence of ALRI, mean (SD)	Number of new cases in a year	0.40 (0.62)

**Table 2 toxins-14-00855-t002:** Cognitive, motor functions, length-for-age-z-scores, morbidity, and aflatoxin exposure of children at 15 months, 24 months, and 36 months of age.

Indicator	15 Months (*n* = 194)	24 Months (*n* = 167)	36 Months (*n* = 163)
Aflatoxin detected, *n* (%)	40 (20.6%)	28 (16.8%)	99 (60.7%)
LAZ, mean (SD)	−1.80 (0.930)	−2.03 (0.96)	−1.99 (0.84)
Haemoglobin, mean (SD)	11.3 (1.48)	11.7 (1.51)	12.3 (1.33)
Cognitive T-scores, mean (SD)	39.0 (3.14)	50.4 (3.98)	61.9 (3.05)
Receptive T-scores, mean (SD)	39.1 (3.79)	50.8 (3.10)	61.4 (4.56)
Expressive T-scores, mean (SD)	39.0 (4.42)	51.9 (5.71)	60.2 (3.75)
Fine motor T-scores, mean (SD)	40.2 (2.90)	48.7 (3.52)	62.2 (5.68)
Gross motor T-scores, mean (SD)	39.7 (6.55)	50.5 (3.43)	60.7 (4.28)

**Table 3 toxins-14-00855-t003:** Bivariate analyses with nutrition status, cognitive, and motor functions after adjusting for age and sex.

Variable	Cognitive		Receptive		Expressive		Fine Motor		Gross Motor	
	Beta	*p*-Value	Beta	*p*-Value	Beta	*p*-Value	Beta	*p*-Value	Beta	*p*-Value
Aflatoxin detected	−0.702	0.036	−0.844	0.021	−0.927	0.056	0.696	0.078	−0.075	0.872
Birth weight in kg	0.977	0.022	1.529	0.003	1.863	0.002	1.206	0.02	1.988	0.003
Birth order	−0.063	0.676	−0.216	0.234	−0.418	0.052	−0.087	0.634	−0.091	0.699
Length for age z score	0.415	0.022	0.45	0.035	0.591	0.021	0.239	0.279	1.501	<0.001
Haemoglobin	0.112	0.308	0.012	0.922	0.331	0.036	0.07	0.592	0.421	0.008
Incidence of diarrhoea	−0.071	0.377	−0.193	0.046	−0.217	0.059	−0.111	0.257	−0.181	0.151
Incidence of ALRI	−0.045	0.875	−0.105	0.76	−0.494	0.227	0.007	0.983	0.597	0.179
Soluble TfR	0.023	0.687	−0.045	0.501	−0.082	0.301	−0.054	0.417	−0.037	0.675
Maternal age (y)	−0.038	0.271	−0.076	0.068	−0.072	0.148	−0.034	0.416	−0.036	0.5
Maternal education (y)	0.169	0.002	0.263	<0.001	0.343	<0.001	0.191	0.004	0.223	0.008
Maternal height (cm)	0.073	0.041	0.081	0.06	0.135	0.007	0.079	0.068	0.144	0.009
Raven’s factor	0.07	<0.001	0.123	<0.001	0.091	0.001	0.083	0.001	0.115	<0.001
HOME emotional and verbal responsivity score	0.161	0.239	0.203	0.211	0.263	0.174	−0.093	0.575	0.226	0.277
HOME environmental safety score	0.458	0.091	0.622	0.054	0.913	0.017	0.345	0.297	0.908	0.027
HOME cleanliness score	2.703	0.135	1.042	0.627	1.87	0.468	4.354	0.047	1	0.715
SRQ-20 score	−0.019	0.577	0.01	0.802	−0.036	0.461	−0.006	0.868	0.018	0.719
Monthly household income (USD)	0.004	0.018	0.005	0.025	0.008	<0.001	0.002	0.327	0.004	0.156

**Table 4 toxins-14-00855-t004:** Results of the multivariate mixed effect linear regression models showing the mean effects and 95% confidence intervals to explore aflatoxin exposure and child development aged between 15 months and 36 months.

	Cognitive	Receptive	Expressive	Fine Motor	Gross Motor
Aflatoxin detected	−0.69 (−1.36, −0.02) *	−0.90 (−1.62, −0.17) *	−1.01 (−1.96, −0.05) *	0.59 (−0.19, 1.37)	−0.15 (−1.06, 0.77)
Age in months	1.10 (1.06, 1.13) *	1.07 (1.03, 1.11) *	1.01 (0.96, 1.06) *	1.04 (1.00, 1.08) *	0.98 (0.93, 1.03) *
Female sex	0.17 (−0.57, 0.91)	0.27 (−0.58, 1.12)	0.75 (−0.24, 1.73)	0.75 (−0.15, 1.64)	−0.81 (−1.89, 0.28)
Birth weight in kg	0.68 (−0.27, 1.63)	1.17 (0.07, 2.27) *	1.82 (0.54, 3.09) *	0.76 (−0.33, 1.86)	0.24 (−1.13, 1.62)
Birth order			−0.49 (−1.06, 0.09)		
Length for age z score	0.07 (−0.37, 0.51)	−0.17 (−0.67, 0.34)	−0.13 (−0.74, 0.48)		1.07 (0.43, 1.70) *
Haemoglobin			0.23 (−0.08, 0.54)		0.29 (−0.02, 0.60)
Incidence of diarrhoea		−0.15 (−0.35, 0.04)	−0.16 (−0.38, 0.07)		−0.13 (−0.38, 0.12)
Incidence of ALRI					0.88 (−0.03, 1.80)
Maternal age (y)		−0.06 (−0.15, 0.02)	0.03 (−0.11, 0.16)		
Maternal education (y)	0.06 (−0.07, 0.19)	0.03 (−0.13, 0.18)	0.16 (−0.02, 0.33)	0.07 (−0.08, 0.23)	−0.04 (−0.23, 0.15)
Maternal height (cm)	0.03 (−0.05, 0.10)	0.02 (−0.07, 0.10)	0.05 (−0.06, 0.15)	0.03 (−0.05, 0.12)	0.04 (−0.07, 0.16)
Raven’s factor	0.05 (0.00, 0.10) *	0.11 (0.05, 0.16) *	0.02 (−0.04, 0.09)	0.06 (−0.00, 0.11)	0.07 (0.00, 0.14) *
HOME emotional and verbal responsivity score			0.36 (−0.04, 0.76)		
HOME environmental safety score	0.19 (−0.40, 0.78)	0.20 (−0.48, 0.88)	0.52 (−0.29, 1.32)		0.28 (−0.58, 1.14)
HOME cleanliness score	2.99 (−1.05, 7.04)			5.11 (0.30, 9.92) *	
Monthly household income (USD)	0.00 (−0.00, 0.01)	0.00 (−0.00, 0.01)	0.00 (−0.00, 0.01)		0.00 (−0.00, 0.01)

** p* < 0.05; ALRI: Acute lower respiratory infection; HOME: Home Observation for Measurement of the Environment; WAMI: Water and sanitation, Assets, Maternal education, and Income score.

## Data Availability

Data related to this manuscript are available upon request and for researchers who meet the criteria for access to confidential data may contact with Armana Ahmed (gro.brddci@anamra) of the Research Administration of icddr,b (http://www.icddrb.org/).

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
