# Peer review of "Chronic Aflatoxin Exposure and Cognitive and Language Development in Young Children of Bangladesh: A Longitudinal Study"

_toxins, 2022, doi:10.3390/toxins14120855_

Round 1
Reviewer 1 Report
The topic is very interesting and within the scope of this Journal.
Introduction
Line 25: Aspergillus parasiticus should be written corrrectly.
Line 85: Please, the MAL-ED concept should be explained.
Materials and Methods:
It is not clear the number of children which are included in the study. It would be interesting to add the number of children in each group of age.
Line 93: Please specify the region of the study.
Line 116: Please complete the explanation of “Complementary feeding”. It .should be interesting to include the ingredients
Line 124: Please specify the number of biological samples including plasma and fecal samples.
Line 129: Write “quantitated” in other way
Line 197: It is better to include WHO meanig before (Line 182).
Results
Line 254: These results are not clear. I don´t know the units you used in this table. Maybe you can add more columns.
Author Response
Reviewer-1: Comments and Suggestions for Authors
The topic is very interesting and within the scope of this Journal.
Response: Thank you very much.
Introduction
Line 25: Aspergillus parasiticus should be written correctly.
Response: We have revised it accordingly.
Line 85: Please, the MAL-ED concept should be explained.
Response: We have explained the MAL-ED study.
Materials and Methods:
It is not clear the number of children which are included in the study. It would be interesting to add the number of children in each group of age.
Response: Since the number of children participants in each group is under the result section. We have included this in the result section. Please see page 2 lines 92-94.
Line 93: Please specify the region of the study.
Response: We have included the region of the study. Please see page 2 line 93.
Line 116: Please complete the explanation of “Complementary feeding”. It .should be interesting to include the ingredients
Response: Thank you very much for the comment. Under the data collection section in the methodology, we have mentioned that we have collected data on infant feeding status including data on complementary feeding. Please note that this is the 3rd paper under this aflatoxin study. In the first paper, we explored the association between aflatoxin status and dietary consumption of children using 24-hour recall data. This paper has been published in the Journal of Exposure Science and Environmental Epidemiology (doi: 10.1038/s41370-018-0066-5). The objective of the current manuscript was to explore the association between aflatoxin exposure and cognitive and language development indicators of children. Therefore, we have not included this information.
Line 124: Please specify the number of biological samples including plasma and fecal samples.
Response: Thank you very much. The current analysis did not use any data from fecal sample analysis. Therefore, we have removed this from the method section. We have collected plasma samples for aflatoxin assay to detect aflatoxin exposure for 194, 167, and, 163 Bangladeshi children enrolled in the MAL-ED aflatoxin study at 15, 24, and 36 months of age, respectively. Please see page 2, line 93.
Line 129: Write “quantitated” in other way
Response: We have revised the sentence. Please see page 10, line 290.
Line 197: It is better to include WHO meaning before (Line 182).
Response: We have elaborated the meaning of WHO. Please see page 11, line 342.
Results
Line 254: These results are not clear. I don´t know the units you used in this table. Maybe you can add more columns.
Response: Thank you very much. We have revised Table 1 and a new column is now included to provide units.

Reviewer 2 Report
Kindly find attached a pdf with some minor comments.
Overall a well written and very insightful manuscript.
Introduction: for more clarity provide the specific model that was used to investigate the effect of AFB on the brain, such as when an animal or in vitro model was used.
Methods/Results / Discussion: why as infant feeding practices not measured?

Author Response
Reviewer-2: Comments and Suggestions for Authors
Kindly find attached a pdf with some minor comments.
Response: We have addressed all the comments from the pdf. Please see below.
Introduction: We have included reference for the human study (page 2, line 67).
Methods: We have changed the word “slum” and used appropriate euphemisms (page 10).
Overall a well-written and very insightful manuscript.
Response: Thank you very much.
Introduction: for more clarity provide the specific model that was used to investigate the effect of AFB on the brain, such as when an animal or in vitro model was used.
Response: We have included a new sentence on this. Please see page 2 line 74.
Methods/Results / Discussion: why as infant feeding practices not measured?
Response: We have collected data on infant feeding practices. The scope of the current analysis is to examine the association between aflatoxin exposure and cognitive and language development in children. From this Aflatoxin study, we have already published one paper where we examined the association between infant feeding and aflatoxin exposure. Please see below the reference of this paper.
Mahfuz M, Alam MA, Fahim SM, Gazi MA, Raihan MJ, Hossain M, Egner PA, et al. Aflatoxin exposure in children living in Mirpur, Dhaka: data from MAL-ED companion study. J Expo Sci Environ Epidemiol. 2018 Sep 5. doi: 10.1038/s41370-018-0066-5.
